# Dynamic response and low voltage ride-through enhancement of brushless double-fed induction generator using Salp swarm optimization algorithm

Ahsanullah Memon[1,2]*, Mohd Wazir Bin Mustafa[1], Waqas Anjum[1,3], Ahsan Ahmed[4], Shafi Ullah[5], Saleh Masoud Abdallah Altbawi[1], Touqeer Ahmed Jumani[2], Ilyas Khan[6], Nawaf N. Hamadneh[7]

1 School of Electrical Engineering, Faculty of Electrical Engineering, Universiti Teknologi Malaysia (UTM), Skudai, Johor, Malaysia, 2 Department of Electrical Engineering, Mehran University of Engineering and Technology, SZAB Campus, Khairpur Mir's, Sindh, Pakistan, 3 Department of Electronic Engineering, The Islamia University of Bahawalpur, Bahawalpur, Pakistan, 4 Department of Information Technology, College of Computer and Information Sciences, Majmaah University, Al-Majmaah, Saudi Arabia, 5 Department of Computer Engineering, Faculty of Information and Communication Technology, BUITEMS, Quetta, Pakistan, 6 Department of Mathematics, College of Science Al-Zulfi, Majmaah University, Al-Majmaah, Saudi Arabia, 7 Department of Basic Sciences, College of Science and Theoretical Studies, Saudi Electronic University, Riyadh, Saudi Arabia

* memon.ahsanullah@graduate.utm.my

**Data Availability Statement:** All relevant data are within the paper.

## Abstract

A brushless double-fed induction generator (BDFIG) has shown tremendous success in wind turbines due to its robust brushless design, smooth operation, and variable speed characteristics. However, the research regarding controlling of machine during low voltage ride through (LVRT) need greater attention as it may cause total disconnection of machine. In addition, the BDFIG based wind turbines must be capable of providing controlled amount of reactive power to the grid as per modern grid code requirements. Also, a suitable dynamic response of machine during both normal and fault conditions needs to be ensured. This paper, as such, attempts to provide reactive power to the grid by analytically calculating the decaying flux and developing a rotor side converter control scheme accordingly. Furthermore, the dynamic response and LVRT capability of the BDFIG is enhanced by using one of the very intelligent optimization algorithms called the Salp Swarm Algorithm (SSA). To prove the efficacy of the proposed control scheme, its performance is compared with that of the particle swan optimization (PSO) based controller in terms of limiting the fault current, regulating active and reactive power, and maintaining the stable operation of the power system under identical operating conditions. The simulation results show that the proposed control scheme significantly improves the dynamic response and LVRT capability of the developed BDFIG based wind energy conversion system; thus proves its essence and efficacy.

**Funding:** The author(s) received no specific funding for this work. NO SOURCE OF FUNDING. NEED DISCOUNT.

**Competing interests:** The authors have declared that no competing interests exist.

# 1. Introduction

The need for electrical energy is increasing due to the rapidly increasing population and users' power demand. This continuous increase in power consumption creates several challenges for the electric power companies such as overloading of the existing generating units, transmission lines, transformers, and feeders [1]. Furthermore, the environmental and economic impact of such power systems made them outdated for electricity generation. It is for the mentioned reasons that the trend of relying on fossil fuel-based centralized power systems is declining worldwide and decentralized green energy sources such as wind, solar, and sea tides are considered as potential candidates for power generation. Among the stated green energy options, wind energy has shown quite promising results in terms of reliability and efficiency. However, it has been observed that the increasing penetration of wind energy into the existing power system has introduced few new challenges in terms of grid codes for the advanced generating units, power electronic interfacing devices, and control strategies [2, 3]. In most wind turbines, Double-Fed Induction Generator (DFIG) is utilized to convert the mechanical energy, acquired from the wind, into electrical energy due to its robust design and smooth operation. However, DFIG transfers the generated power to the load by using carbon brushes and slip rings which causes power loss and sparking due to continuous mechanical wear and tear in the mentioned devices [4]. Besides, it provides an inferior low voltage ride through (LVRT) capability as compared to Brushless DFIG which is a comparatively better alternative to the traditional DFIG [5].

Low voltage ride-through, also referred to as fault ride-through, represents the capability of electrical energy generators to remain connected to the power grid during voltage dips [6]. The main motivation behind using generators with good LVRT capability is to prevent widespread loss of generation caused by a short circuit at high voltage (HV) or extremely high voltage (EHV) levels [7]. The winding currents in a generator are responsible to maintain the magnetic field with minimum voltage for its normal operation [8]. However, if this voltage drops below a certain level, the generator may disconnect from the power grid leading to unintentional blackouts [9]. This problem becomes even severe in the case of BDFIG based power system due to the presence of two highly inductive power windings. To avoid the mentioned issue, the researchers have explored several hardware and software-based control strategies. The most widely utilized LVRT enhancement scheme in literature is the Crowbar approach [10, 11]. In this LVRT enhancement approach, a three-phase resistive circuit is placed in series with the CW of the BDFIG to protect the power electronic-based machine side and grid side converters during fault condition without disconnecting the generator from the grid [12]. Despite the very simple operating mechanism and implementation, this approach makes the machine consume a huge amount of power in the resistive circuit during a fault condition and will absorb the reactive power from the grid which consequently leads to unnecessary electromagnetic torque oscillations in the system. Furthermore, the additional resistive circuit requires an extra cost and maintenance which makes it an inefficient and costlier LVRT enhancement option. On the other hand, the software-based LVRT enhancement methods utilize different control schemes to protect the converters during fault conditions [13]. Research studies that have been conducted to determine the effect of control techniques on LVRT capabilities of brushless doubly-fed induction generators include the following. Using an analytical model, Shao et al. [14, 15] introduced an LVRT scheme for symmetric and asymmetric voltage dips based on fixed values for the flux value. [13] is another article that proposes a similar approach to this. According to [14, 15], LVRT can be implemented using positive and negative sequence currents. Other works include [8, 16–18], However, all of these techniques have a similar problem in that the LVRT capability is provided by using fixed reference values for

decaying flux. The flux during fault is responsible for the production of electro motive force which in result increase's fault current and may damage the power electronics converters. Therefore, the flux should be reduced to zero instantly. Moreover, above studies provide poor dynamic response during normal and fault conditions.

To solve the first problem identified above, i.e., use of fixed control flux reference values for reactive current control, in this paper, the reference value for flux is analyzed using an analytical model in the event of a short-circuit fault. The brushless doubly-fed induction generator's equivalent circuit is used to analyze the load's reactive current steady state and transient state components. This leads to a dynamic reference value for the control winding current obtained from the load current. Thus, the purpose of this paper is to develop an rotor side converter control scheme for a brushless doubly-fed induction generator that will improve the LVRT characteristics during faults.

All the referred studies have utilized the vector control scheme with manual PI tuning through the "trial and error" method for accomplishing the corresponding control tasks. The quoted PI tuning method is time-consuming and inefficient and does not guarantee the optimal selection of gains at the end of the tuning process; thus, affects the overall performance of the control architecture in terms of managing a suitable dynamics response of the system. To overcome the second stated issue and to enhance the dynamic response of the BDFIG, the authors in reference [19] have utilized an internal model control (IMC) technique to select the optimal PI regulator gains in a vector control scheme. The IMC is an analytical method for extracting the optimal PI gains from the detailed extensive model of the studied control system. Since most of the wind energy conversion systems consists of very complex power and control circuits; therefore, it's very hard to derive an accurate mathematical of model of such systems. One of the effective methods to overcome this problem is to utilize a PI tuning approach that does not require such extensive modeling of the system and may treat the studied system as black box. One of such approaches is to select the PI gains with metaheuristics-based optimization methods. The operation of these optimization methods is orientated on minimizing or maximizing a pre-defined FF through a stochastic based searching mechanism [20]. A Grouped grey wolf optimizer for doubly-fed induction generator [21], Democratic joint operations algorithm [22], Adaptive fractional-order PID control for Permanent magnet synchronous generator (PMSG) [23] and grouped grey wolf optimizer for a grid-connected PV inverter [24] have been used to achieve maximum power point tracking (MPPT) in mentioned machine controls.

This paper explores the application of one of the most intelligent metaheuristic optimization algorithms called the Salp Swarm Algorithm (SSA) in achieving the optimal dynamic response and LVRT capability enhancement of a grid-tied BDFIG system during normal and fault conditions. The SSA is utilized to select the most optimal PI regulator gains in the proposed vector control scheme. One of the significant aspects of this research work is that the current work utilizes Grid Side Converter (GSC) in conjunction with the Machine Side Converter (MSC) for enhancing the reactive power support to the grid in order to avoid any disconnection during the fault condition. The dynamic response obtained from the SSA-based control scheme is evaluated and compared with the Particle Swarm Optimization (PSO) algorithm-based controller in terms of limiting the fault current, regulating dc bus voltage, and active and reactive power under identical operating conditions. The proposed technique ensures injection of controlled reactive power to the grid during fault conditions to avoid grid disconnection; thus, provides the stable operation of the machine without any usage of hardware such as a crowbar. The major contributions of this article are listed as follows.

1. An extensive analysis of flux variation during normal and fault condition and developing RSC control scheme.

2. To improve the dynamic response of the considered WECS under normal and fault operating conditions, the PI regulators' gains are optimized automatically using the intelligence of SSA.

3. A comprehensive comparative analysis of the proposed control scheme with IMC and PSO-based VC schemes is made based on few important dynamic response evaluation metrics such as parentage overshoot and settling time to validate the efficacy and essence of the proposed WECS control architecture.

The rest of the paper is organized as follows: Section 2 presents the considered BDFIG and its mathematical foundation. The details of the proposed control scheme are provided in Section 3. Simulation results are discussed in Section 4 while the conclusion for the current research work is laid in Section 5.

## 2. Modeling of vector control scheme based BDFIG system

This section is further divided into two subsections.

### 2.1 An analytical model of BDFIG

The BDFIG consists of two windings i.e., Power winding (PW) and Control Winding (CW). The generated power from BDFIG is directly transferred to the grid through PW while the Control winding (CW) is connected to the grid through a power electronic interface [25]. To enable vector control of the machine, the PW and CW voltages are translated into the $dq$ reference frame which is aligned with the PW flux orientation in the current case. The resulting equations for PW, CW and Rotor Windings (RW) are depicted as follows [26].

$$V_p = I_p R_{sp} - j\omega_p \lambda_p + \frac{d\lambda_p}{dt} \tag{1}$$

$$V_c = I_c R_{sc} - j(\omega_p - N_r \omega_r)\lambda_c + \frac{d\lambda_c}{dt} \tag{2}$$

$$V_r = 0 = I_r R_r - j(\omega_p - P_p \omega_r)\lambda_r + \frac{d\lambda_r}{dt} \tag{3}$$

$$\lambda_p = L_{sp} I_p + L_{hp} I_{rd} \tag{4}$$

$$\lambda_c = L_{sc} I_c + L_{hc} I_r \tag{5}$$

$$\lambda_r = L_{hp} I_p + L_{hc} I_c + L_r I_r$$

where $V_p$, $V_c$ and $V_p$ denote the PW, CW and RW voltage while $\lambda_p$, $\lambda_c$ and $\lambda_r$ denote the PW, CW and RW flux. Finally, the rotor torque can be calculated as follows;

$$T_e = -\frac{3}{2} P_p I_m [\lambda_p^* I_p] - \frac{3}{2} P_c I_m [\lambda_c^* I_c] \tag{6}$$

To control the behavior of the modeled BDFIG based WECS, an SSA-based VC scheme is proposed; the details of which are provided in the subsequent subsection.

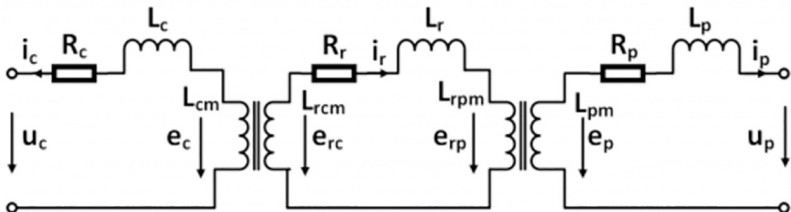

**Fig 1. BDFIG equivalent circuit.**

## 2.2 Proposed technique for RSC

The per-phase steady-state equivalent circuit of a BDFIG is shown in Fig 1. The RSC and GSC are decoupled using the capacitance $C_{dc}$ therefore, the analysis for them can be done independently.

To analyses the situation of LVRT during a short circuit fault, the single-phase equivalent circuit for the stand-alone BDFIG system seen from the CW side is shown in Fig 2, $i_{La}$ is the a-phase load current, and $i_g$ represents the short circuit fault current. $u_a$ and $i_a$ are the voltage and current for the a-phase at the PCC. Denoting the amplitude and angular frequency of the phase voltage by $U_m$ and $\omega$, respectively, we can obtain $e_{ca}$ as follows [5].

$$e_{ca}(t) = U_m cos(\omega t) \tag{7}$$

Taking the Laplace transform of Eq (7), we obtain $e_{ca}$ in complex frequency domain as follows:

$$e_{ca}(s) = \frac{U_m s}{s^2 + \omega^2} \tag{8}$$

Similarly, for the current source representing the short circuit fault can be written in the frequency domain as:

$$e_s(s) = \frac{U_s s}{s^2 + \omega^2} \tag{9}$$

where $U_s$ is the amplitude of the voltage after the short circuit fault. At natural speed, $i_a = 0$ and under normal conditions, it can be assumed without loss of generality that the grid line impedance is negligible. In the presence of a short circuit fault, the switch in Fig 2 will be

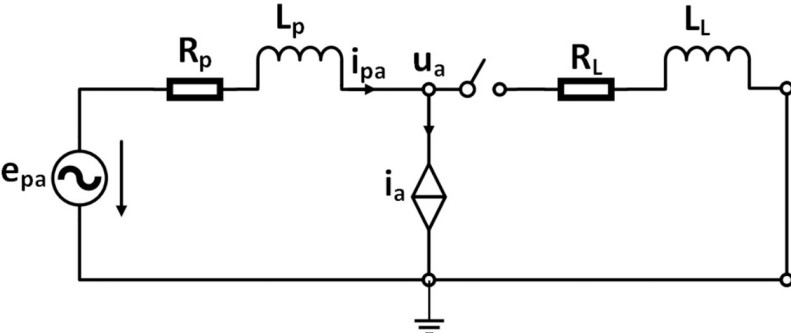

**Fig 2. Single-phase equivalent circuit from PW.**

closed and the stator CW current can be written as:

$$i_{ca}(s) = \frac{e_{ca}(s) - e_s(s)}{L_s + R} = \frac{U}{(L_s + R)(s^2 + \omega^2)} \tag{10}$$

At natural speed, $i_a = 0$ and under normal conditions, it can be assumed without loss of generality that the grid line impedance is negligible. When a load is connected, the switch in Fig 2 will be closed and the stator PW current can be written as:

$$i_{pa}(s) = U_m s \frac{e_{ca}(s)}{L_s + R} = \frac{U_m}{(L_s + R)(s^2 + \omega^2)} \tag{11}$$

where, the total inductance $L$, and total resistance $R$ are given by

$$R = R_c + R_l + R_s \tag{12}$$

$$L = L_c + L_l + L_L \tag{13}$$

$$U = U_m - U_s \tag{14}$$

Defining the magnitude and angle of the total impedance as follows:

$$|Z| = \sqrt{R^2 + (\omega L)^2} \tag{15}$$

$$\phi = \arctan\left(\frac{\omega L}{R}\right) \tag{16}$$

Then Eq (11) can be re-written as [27]:

$$i_{ca}(s) = \frac{U}{|Z|^2}\left(\frac{R_s}{(s^2 + \omega^2)} + \frac{\omega^2 L}{(s^2 + \omega^2)} - \frac{RL}{L_s + R}\right) \tag{17}$$

$$i_{ca}(s) = \frac{U}{|Z|}\left(\frac{s\cos\phi}{(s^2 + \omega^2)} + \frac{\omega\sin\phi}{(s^2 + \omega^2)} - \frac{\cos\phi}{L_s + R}\right) \tag{18}$$

Taking the inverse Laplace transform, we get:

$$i_{ca}(s) = \frac{U}{|Z|}\cos(\omega t - \phi) - \frac{U}{|Z|}\cos(-\phi)e^{-\frac{R}{L}t} \tag{19}$$

Eq (19) shows that the CW current is composed of two components: first, a fundamental frequency component, and second, an exponentially decaying DC component. The equations for b and c phase currents of the stator CW can be obtained similarly. Thus, the stator CW current consists of a steady state component and transient component. The stator CW current in the synchronous reference frame is obtained using Park transformation:

$$i_{cd}(t) = \frac{U}{|Z|}\cos(\phi) - \frac{U_m}{|Z|}\cos(\omega t + \phi)e^{-\frac{R}{L}t} \tag{20}$$

$$i_{cq}(t) = -\frac{U}{|Z|}\sin(\phi) + \frac{U_m}{|Z|}\sin(\omega t + \phi)e^{-\frac{R}{L}t} \tag{21}$$

where the steady state and transient state components can be separated as follows:

$$i_{cds}(t) = \frac{U_m}{|Z|}\cos(\phi) \;\; i_{cdt} = \frac{U_m}{|Z|}\cos(\omega t + \phi)e^{-\frac{R}{L}t} \tag{22}$$

$$i_{cqs}(t) = -\frac{U_m}{|Z|}sin(\phi) \;\; i_{cdt} = \frac{U_m}{|Z|}\sin(\omega t + \phi)e^{-\frac{R}{L}t} \tag{23}$$

Thus, the active and reactive CW currents are also composed of a DC steady state component and an exponentially decaying transient component. From above equation flux value is calculated.

### 2.3 Vector control scheme

The block diagram of the proposed grid-connected BDFIG based vector control scheme is shown in Fig 1.

In the vector control scheme shown in Fig 3, the MSC is directly connected to the CW and is responsible for controlling the active and reactive power with q and d components of CW current respectively [28, 29]. A detailed inside structure of the MSC and GSC are depicted in Figs 4 and 5 respectively.

As can be seen from Fig 4, the Maximum Power Point Tracking (MPPT) is utilized to extract the maximum available power from the wind energy which is then utilized as the reference active power to be attained from the considered BDFIG based wind turbine. The wind speed $Vw$ is considered as constant (12m/s) throughout the simulation run. To accurately track the reference reactive power with minimum error, four PI regulators with optimal gains are utilized in MSC. The gains of these PI regulators are optimized through SSA to achieve the optimal dynamic response of the system during normal and fault conditions.

Another component of the proposed vector control scheme is GSC as shown in Fig 5. Instantly after the fault occurs, the dc-link voltage overshoots which consequently causes an undesirable operating condition of the BDFIG based power system [16, 30]. To avoid such conditions and to enhance the reactive power support to the grid during fault occurrence, a

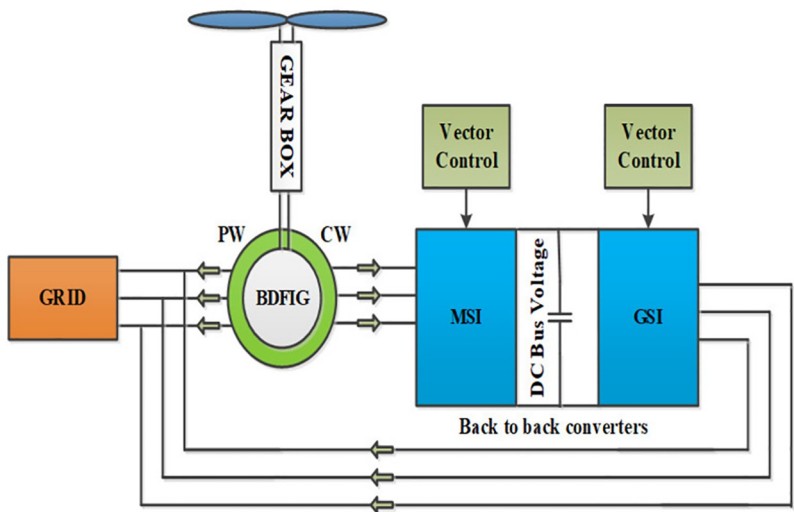

**Fig 3. BDFIG-WECS system.**

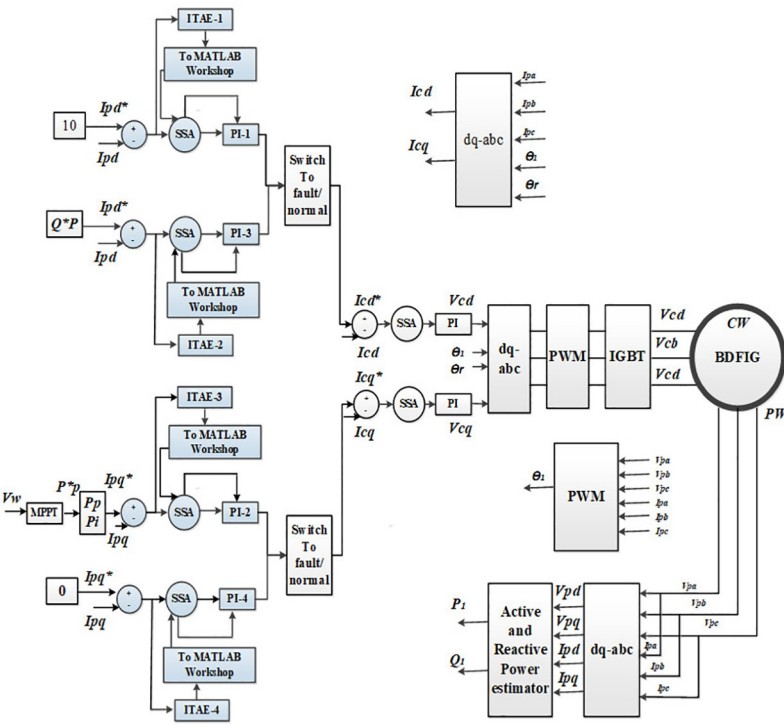

**Fig 4. Machine side converter.**

grid side converter is introduced in the proposed control scheme. The GSC is used to control the dc-link voltage and reactive power whose reference value is set as zero. In typical power grids, a voltage dip may cause a huge deviation in critical system parameters such as active/ reactive power, torque, and dc-link voltage. To solve this issue, a coordinated control strategy for the MSC and GSC is explored in current research work. Using a similar approach as that of MSC, the GSC controller is designed with PI tuning through SSA; thus, optimal dynamic response of the proposed system is achieved.

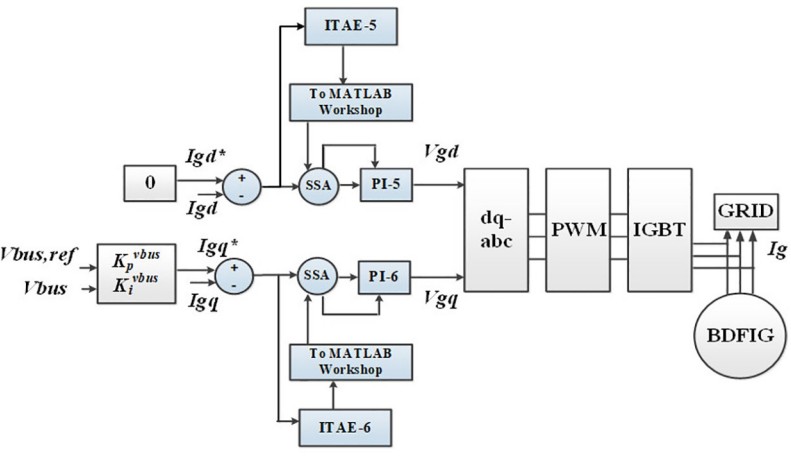

**Fig 5. Grid side converter.**

## 3. Proposed optimization method

This section provides the detailed working mechanism and implementation of the SSA in solving the current optimization problem. Owing to the simple working mechanism, the capability of avoiding stagnation into the local minimum, and good exploration versus exploitation characteristics, the SSA is chosen as the optimization method for selecting the most suitable PI gains in the current study. This is achieved by minimizing an error integrating fitness function through the intelligence of mentioned algorithm. The function opted for the current study is provided in Eq (24).

$$FF = \int_0^t t|e_1(t)| \, dt + \int_0^t t|e_2(t)| \, dt + \int_0^t t|e_3(t)| \, dt + \int_0^t t|e_4(t)| \, dt + \int_0^t t|e_5(t)| \, dt + \int_0^t t|e_6(t)| \, dt \quad (24)$$

where $t$ is the simulation time while $|e_1(t)|$ to $|e_6(t)|$ represents the corresponding error signals for the control loops depicted in Figs 4 and 5. The detailed working mechanism of the SSA is provided in the subsequent subsection which is followed by its implementation in the current study.

### 3.1 Salp swarm optimization algorithm

Salps belong to the family member of the Salpidae group with a translucent barrel-shaped body. To model their movement behavior mathematically, their initial positions are initialized randomly as portrayed in Eq (25) [31].

$$K_1^{1:n} = ran(\ldots). * (ub_j - lb_j) + lb_j, \, \forall j \in no. \, of \, variables \quad (25)$$

where; $K_1^{1:n}$ represents the initial positions of the salps with $ub_j$ and $lb_j$ as their upper and lower limit respectively while the rand denotes the random values between 0 and 1. Once the search agents (salps) are spread over the pre-defined search space randomly, the swarm is divided into two equal groups i.e., "leaders" and "followers". This division is based on the evaluation of the pre-defined fitness function. The top half salps in the context of their higher solution quality are considered as leaders while the rest are considered as followers. The solution in evaluating the fitness function for the leading salps is recorded and updated at the end of each iteration using Eq (26).

$$K_j^1 = \begin{cases} M_i + c_1((ub_j - lb_j)c_2 + lb_j); \, c_3 \geq 0.5 \\ M_i - c_1((ub_j - lb_j)c_2 + lb_j); \, c_3 < 0.5 \end{cases} \quad (26)$$

where the symbol shows the jth dimension position of leader salp, symbolizes the jth dimension location of the food source while $c_1$, $c_2$, and $c_3$ represents the random numbers. It is important to note that, SSA has the ability to avoid to the local minimum effectively as compared to conventional optimization methods like GA and PSO because of its adaptive optimization nature. The position of follower salps regularly is updated by the SSA and follower salps start to shift gradually toward leading salps. This helps the SSA to prevent stagnating into local optima. Thus, an optimal or close to the optimal solution is produced by SSA during the optimization process. Moreover, SSA performs the best exploration versus exploitation balancing capability. Eq (26) shows that the leader salp location is upgraded with reference to the food source only. The coefficient $c_1$ is the main character in SSA as it contributes to equalizing the exploitation and exploration is calculated by using Eq (27).

$$c_1 = 2e^{-\left(\frac{4l}{L}\right)^2} \quad (27)$$

where $l$ shows the updated number of iteration while $L$ symbolizes the maximum number of iterations. Utilizing Newton's second law of motion, the upgraded location of follower salp is given as,

$$K_j^i = \frac{1}{2}at^2 + v_0 t \tag{28}$$

where i $\geq$ 2 and $K_j^i$ symbolize the ith positions of follower salp in jth dimension, t denotes the time while $v_0$ represents the velocity of salps at the initial stage of the optimization process, and is generally considered as 0. As the number of iteration replaces the time during the optimization process and to avoid the fractional number between successive iterations the Eq (29) can be re-written as given underneath.

$$K_j^i = \frac{K_j^i + K_j^{i-1}}{2} \tag{29}$$

where i $\geq$ 2 and $K_j^i$ represents the position of ith follower salp in jth dimension.

## 3.2 SSA implementation in current optimization problem

The proposed methodology of obtaining optimal PI parameters for MSC and GSC to enhance the dynamic response of the grid-tied BDFIG system using the optimization technique SSA is represented in the flow-chart shown in Fig 6.

A new development in this research is the use of three different optimization techniques (IMC, PSO, and SSA) in order to enhance the dynamic performance of the considered BDFIG based WECS. IMC is a method used to determine PI parameters that considers the operating parameters of the system in order to compensate its open-loop poles with the zeros of the PI controller, which provides the desired closed-loop bandwidth and time constant. A detailed mathematical description of IMC can be found in reference [19]. In contrast, PSO and SSA are swarm intelligence-based optimization techniques that that provide the optimized solution through an iterative process as they use swarm intelligence. As opposed to the IMC, swarm intelligence-based methods do not require the system to have a complete mathematical model. As their very first stage of operation, these optimization methods make use of random number generation within the bounded search space. As a result, the fitness of each search agent is calculated and their position is updated iteratively based on the governing equations of the algorithm until the specified number of iterations has been reached.

Initially, the code for the SSA is encrypted in MATLAB 2018b editor window while the model for the BDFIG along with the proposed control scheme is designed in Simulink. Before starting the optimization process, the optimization parameters such as the number of iterations, numbers of search agents along with their upper and lower bounds, decision variables, and simulation time is encrypted in SSA code. The aim is to achieve optimal values for the decision variables (PI gains in the current study) by minimizing an error integrating fitness function. Like any other metaheuristic-based optimization methods, the SSA initiates its optimization mechanism by spreading a pre-decided number of search agents in bounded search space. Afterward, the fitness function of each search agent is calculated and sorted according to their magnitudes in descending order. During each iteration, the algorithm compares the best among the current search agents with the best among the previously acquired search agents in terms of the magnitude of the evaluated fitness function. If the current value of the fitness function is less than the previously recorded value of the same, the current fitness function value will be stored as global best while if the mentioned condition is not fulfilled, the algorithm will keep the previous solution as the global best. Hence the most optimal PI

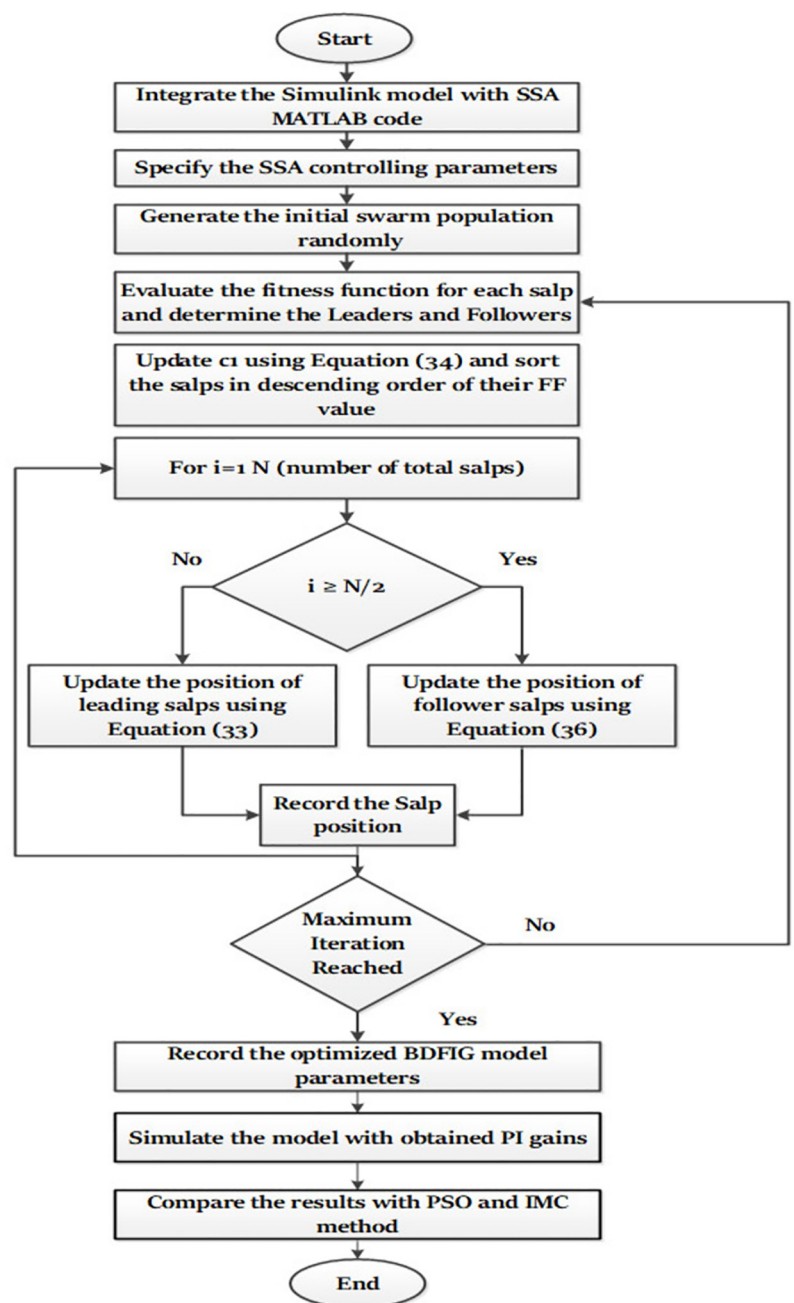

**Fig 6. Flowchart of the proposed methodology.**

regulator gains for both the GSC and MSC are achieved at the end of the simulation that corresponds to the least acquired value of the fitness function. Finally, the model is tested with the obtained PI regulator gains and the corresponding dynamic response is recorded. In order to validate the effectiveness of the proposed control scheme, its dynamic response during normal and fault conditions are compared with that of the PSO-based controller under identical operating conditions and system configuration. The optimization parameters utilized for the PSO and SSA are provided in Table 1.

**Table 1. Optimization parameters.**

| Parameters | PSO | SSA |
|---|---|---|
| Max. number of iterations (i) | 50 | 50 |
| Population size (N) | 50 | 50 |
| Upper boundary of PI gains ($ub_j$) | 10 | 10 |
| Lower boundary of PI gains ($lb_j$) | 0.001 | 0.001 |
| Number of dimensions (D) | 12 | 12 |
| PSO Cognitive constant (C1) | 0–2 | - |
| PSO social constant (C2) | 2–0 | - |
| Inertia weight ($\varpi$) | 0.9–0.4 | - |
| Random numbers (C1, C2) | - | Random [0–1] |

## 4. Results and discussion

To evaluate the performance of the proposed SSA-based BDFIG vector control scheme in achieving the enhanced dynamic response and LVRT capability of the mentioned system, the developed model is simulated using MATLAB/SIMULINK version 2018b. The Discrete Tustin/Backward Euler (TBE) with sampling time of 50 μs is used for simulating the developed BDFIG Simulink model. The optimized PI gains achieved at the end of the optimization process at MSC for all the studied methods are provided in Table 2.

To instigate a fair comparison among the studied methods the proposed BDFIG vector control scheme is simulated with the attained PI regulator parameters under identical operating conditions and system configurations. The machine parameters utilized in the current study are depicted in Table 3.

The effectiveness of the proposed controller is validated under the following performance evaluation cases.

### 4.1 Convergence profile

In this case, the efficacy of the proposed controller in minimizing the opted FF is evaluated and compared with that of the same for the PSO-based controller under identical optimization parameters and conditions. The convergence curve for both PSO and SSA is depicted in Fig 7.

**Table 2. Optimized PI regulator gains.**

| Optimization variables | IMC | PSO | SSA |
|---|---|---|---|
| $k_p^1$ | 0.0021 | 0.0097 | 0.1301 |
| $k_i^1$ | 0.0152 | 0.0094 | 0.0053 |
| $k_p^2$ | 0.1223 | 0.1902 | 0.0071 |
| $k_i^2$ | 0.0143 | 2.0091 | 0.0051 |
| $k_p^3$ | 0.5455 | 1.2164 | 0.8159 |
| $k_i^3$ | 0.8955 | 1.6219 | 1.2314 |
| $k_p^4$ | 4.5531 | 1.2164 | 2.4044 |
| $k_i^4$ | 0.3401 | 3.4562 | 0.9342 |
| $k_p^5$ | 0.6431 | 1.2431 | 0.6421 |
| $k_i^5$ | 1.4531 | 2.8335 | 1.8550 |
| $k_p^6$ | 0.5690 | 0.3467 | 0.3586 |
| $k_i^6$ | 3.2344 | 1.7443 | 0.4643 |

**Table 3. Machine parameters.**

| Parameters | Value | Parameters | Value |
|---|---|---|---|
| Natural Frequency | 52 rad/sec | Power winding resistance ($R_{sp}$) | 2.3 Ω |
| PW pole pairs ($P_p$) | 2 | Control winding resistance ($R_{sc}$) | 4 Ω |
| CW pole pairs $P_c$ | 4 | Rotor winding resistance ($R_r$) | 1.2967e-4 Ω |
| PW Voltage | 240V | Self-inductance of PW ($L_{sp}$) | 0.3498 H |
| CW Voltage | 240V | Self-inductance of CW ($L_{sc}$) | 0.3637H |
| PW Current | 8A | Self-inductance of RW ($L_r$) | 4.4521e-5 H |
| CW Current | 8A | Mutual inductance between PW and RW ($L_{hp}$) | 0.0031 H |
| Rated Torque | 100 N.m | Mutual inductance between CW and RW ($L_{hc}$) | 0.0022 H |
| Grid frequency | 50 hz | Rotor moment of inertia ($J$) | 0.4 kg.m2 |

It can be observed from Fig 7 that the SSA achieves better solution quality than that of the PSO in terms of minimizing the designed FF. convergence rate and solution quality (minimized FF magnitude) are very important quantities that are responsible to inform directly about the effectiveness of a metaheuristic optimization algorithm. The FF magnitude attained by SSA is 22.00104 while that for the PSO is recorded as 22.23161. The SSA has achieved the stated solution quality in 34 iterations while the PSO has achieved a lesser solution quality as compared to SSA in the 35th iteration of the simulation. Therefore, it is concluded that the SSA achieved a better solution quality and convergence rate than that of PSO under identical optimization parameters and system operating conditions; thus, proves its high-quality optimization capabilities.

## 4.2 Dynamic response evaluation

To evaluate the performance of the proposed RSC control scheme under abnormal operating conditions, a fault was injected at 2 to 3 seconds of the simulation run. The corresponding voltage of the system is shown in Fig 8.

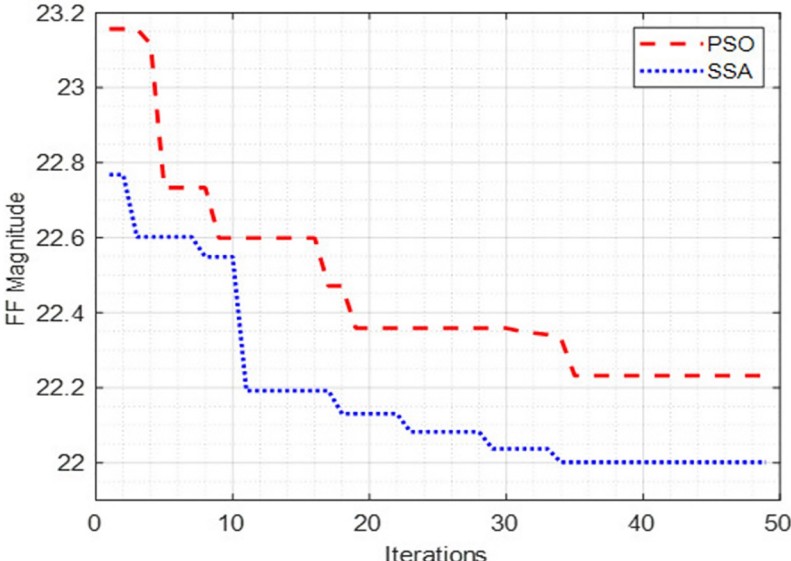

**Fig 7. Convergence curve for PSO and SSA.**

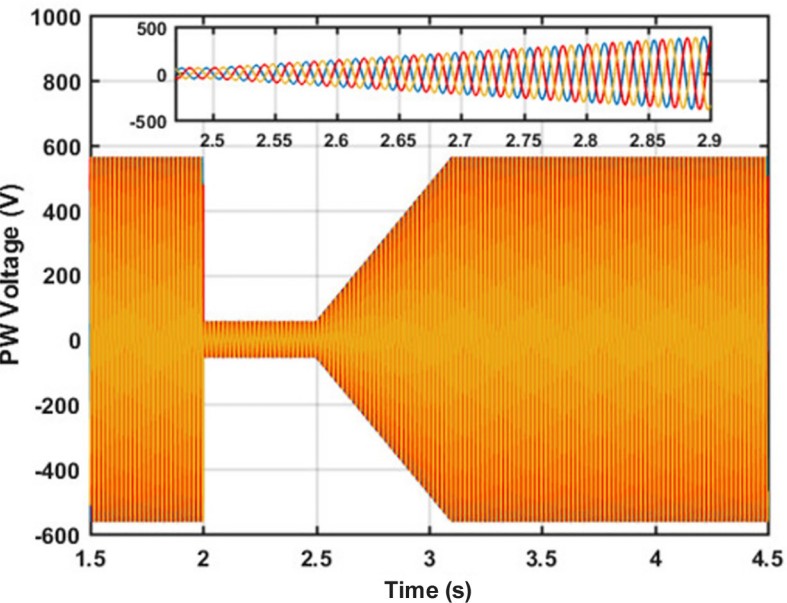

**Fig 8. Voltage waveform during the fault condition.**

As can be seen from Fig 8 that the system works under normal conditions for up to 2 seconds of the simulation run. At exactly 2 seconds, a symmetrical fault is injected into the system and as a result, the system voltage drops by 90%. As the fault is removed, the controller starts to restore the rated voltage value as can be observed from 2.5 to 3.1 seconds in Fig 6. Finally, the rated voltage value is regained after the transient period is over; thus, validates the effectiveness of the proposed control scheme. The analytically calculated flux behavior of the machine has been observed in the simulation during the fault condition period. The PW flux without and with controller is shown in Fig 9(a) and 9(b) respectively.

As can be seen from Fig 9(b) that a dip in the PW flux is observed due to the fault which consequently led to a reduction of the PW flux to zero instantly. The role of PW flux is much important in machines during both normal and fault conditions. It causes to produce emf in rotor which in response to produce higher current and makes the machine windings to burn. As opposed to Fig 9(a), it is important to note that despite the occurrence of a fault the PW flux does not enter the critical region and is successfully recovered to the nominal value by the proposed controller. Furthermore, it can also be observed that the PW flux magnitude is decreasing instantly once the fault is injected, thus minimizing the production of the emf which consequently led to a reduction in current which in turn avoids the heating of the machine and connected equipment. Fig 10(a) shows that the PW voltage and current waveforms are in phase with each other during normal conditions, whereas Fig 10(b) shows that the PW current waveform is lagging behind the voltage waveform during fault conditions; thus, provides the required reactive power to the grid.

In this article, the grid current is controlled to make sure that the machine remains connected with the grid throughout the normal and fault conditions. Fig 11 depicts the efficacy of the proposed control scheme in maintaining the continuity of grid current during the fault condition.

## 4.3 Comparative analysis

Since symmetrical faults in the power grid led to increased machine currents which may cause damage to equipment and loss of capital. For example, an insulated-gate bipolar transistor

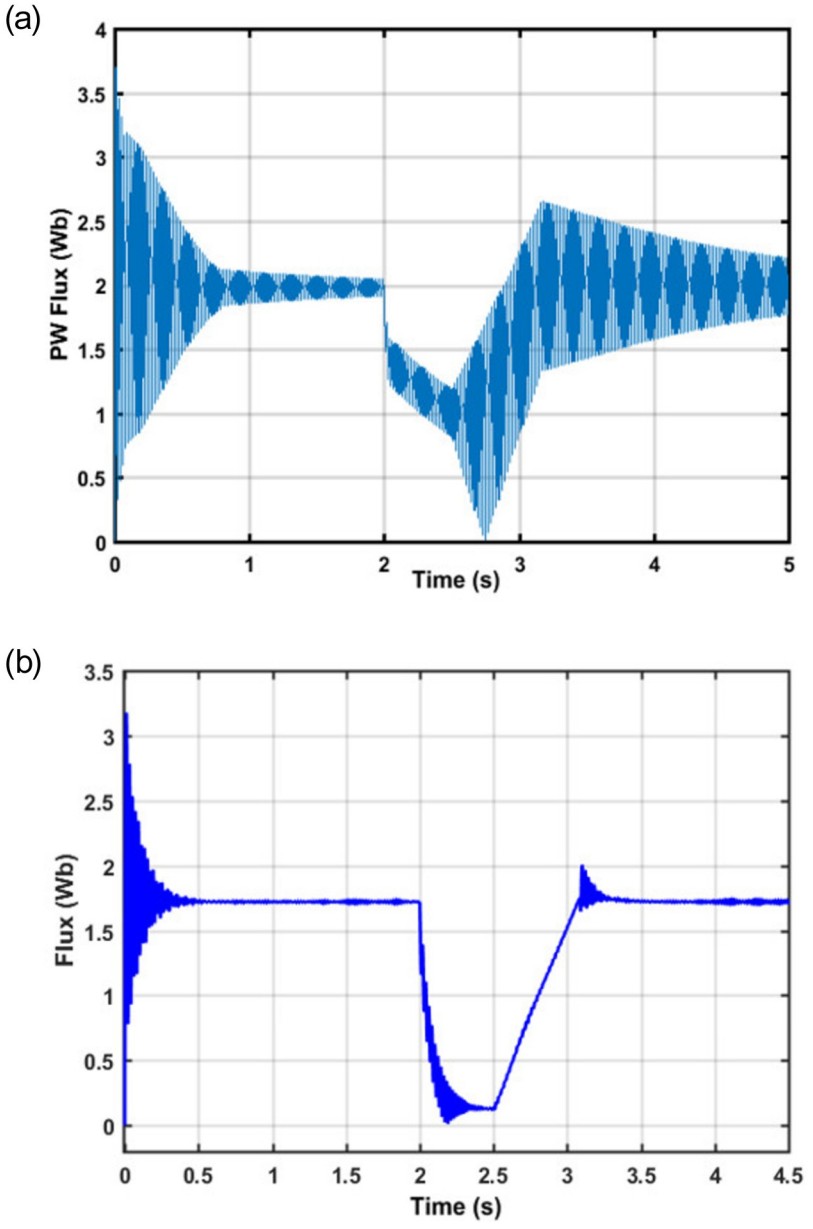

**Fig 9. (a) PW Flux without controller (b) PW Flux with controller.**

(IGBT) can withstand 2 per unit CW current for a 1 ms time period. Therefore, it is important to regulate or divert the fault current to prevent any damage to the connected equipment. The proposed control scheme limits the CW current within the specified tolerance levels. In order to validate the effectiveness of the proposed controller, its performance is compared with the PSO and IMC-based controllers under identical operating conditions and system configurations. The corresponding response of the system in terms of d-q current components for the CW and PW is shown in Figs 12 and 13 respectively.

It can be observed from Figs 12 and 13 that the proposed control scheme regulates the CW and PW current components with the least overshoot and settling time; thus, protects the

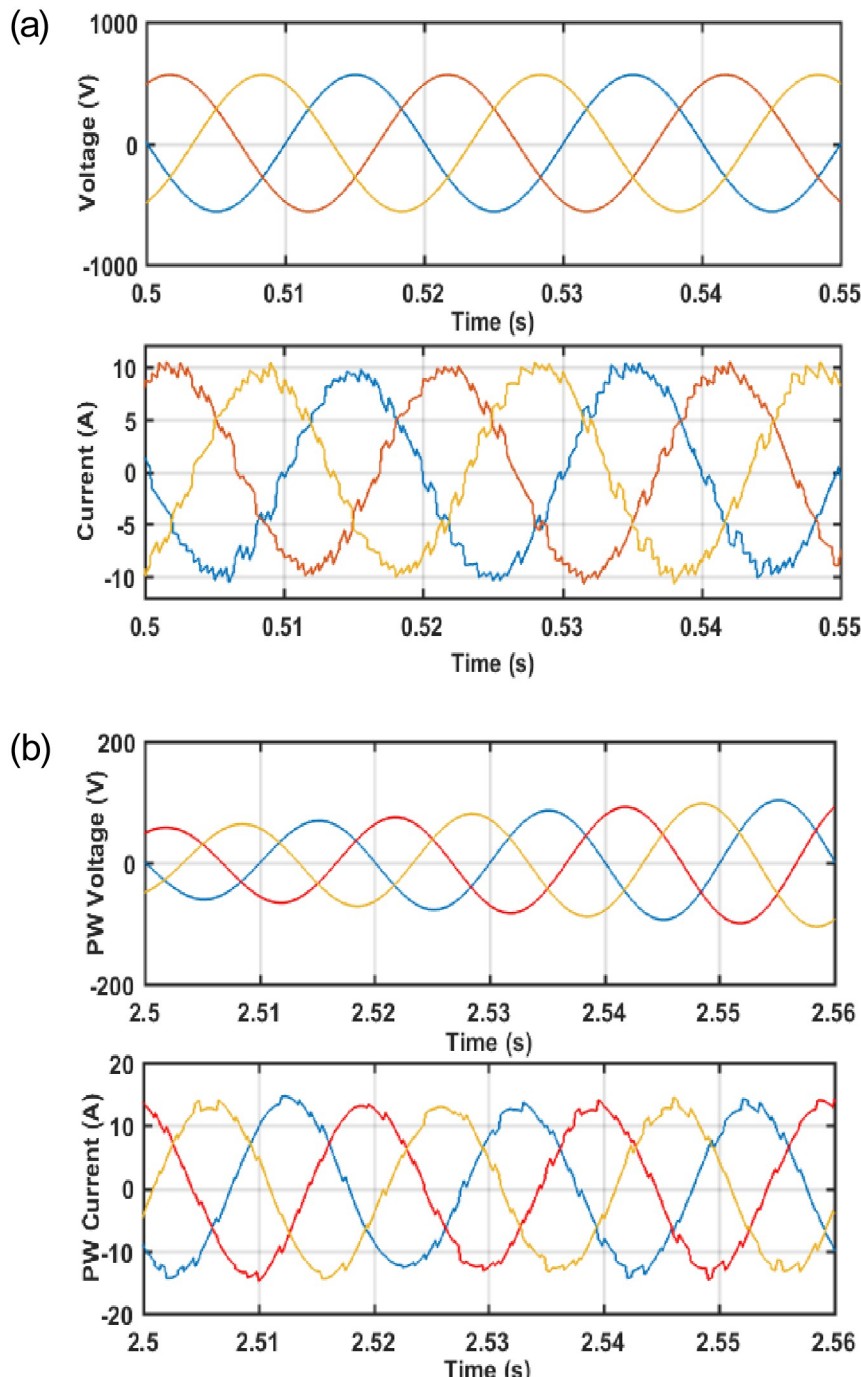

**Fig 10. (a) During normal condition (b) During a fault condition.**

converters from damage due to overcurrent. The proposed controller successfully manages to keep the machine currents within the allowable range even in the presence of a fault at t = 2 seconds. The concerned PW and CW currents return to their pre-fault values after minute sags and swells during the transient period. Besides, it can be observed that the proposed technique results in significantly lowering the oscillations and overshoot in the presence of faults.

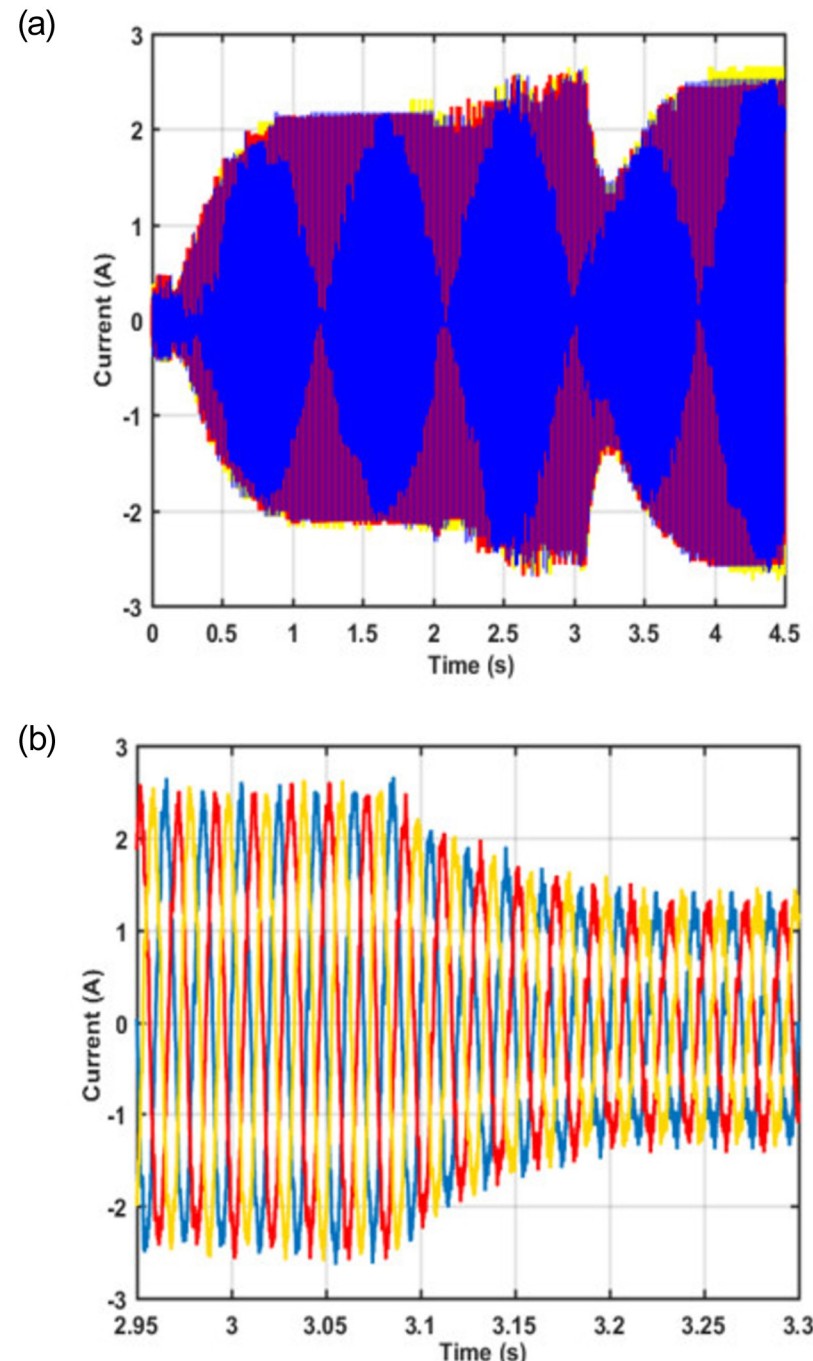

**Fig 11. (a) Grid current (b) zoomed version.**

Tables 4 and 5 show the performance of the SSA in comparison with IMC and PSO for both d and q components of CW and PW currents.

In Tables 4 and 5, the *tr, ts, os, us, and pv* represent the Rise Time, Settling time, Overshoot, Undershoot, and Peak Value respectively. Another very important aspect of any BDFIG based

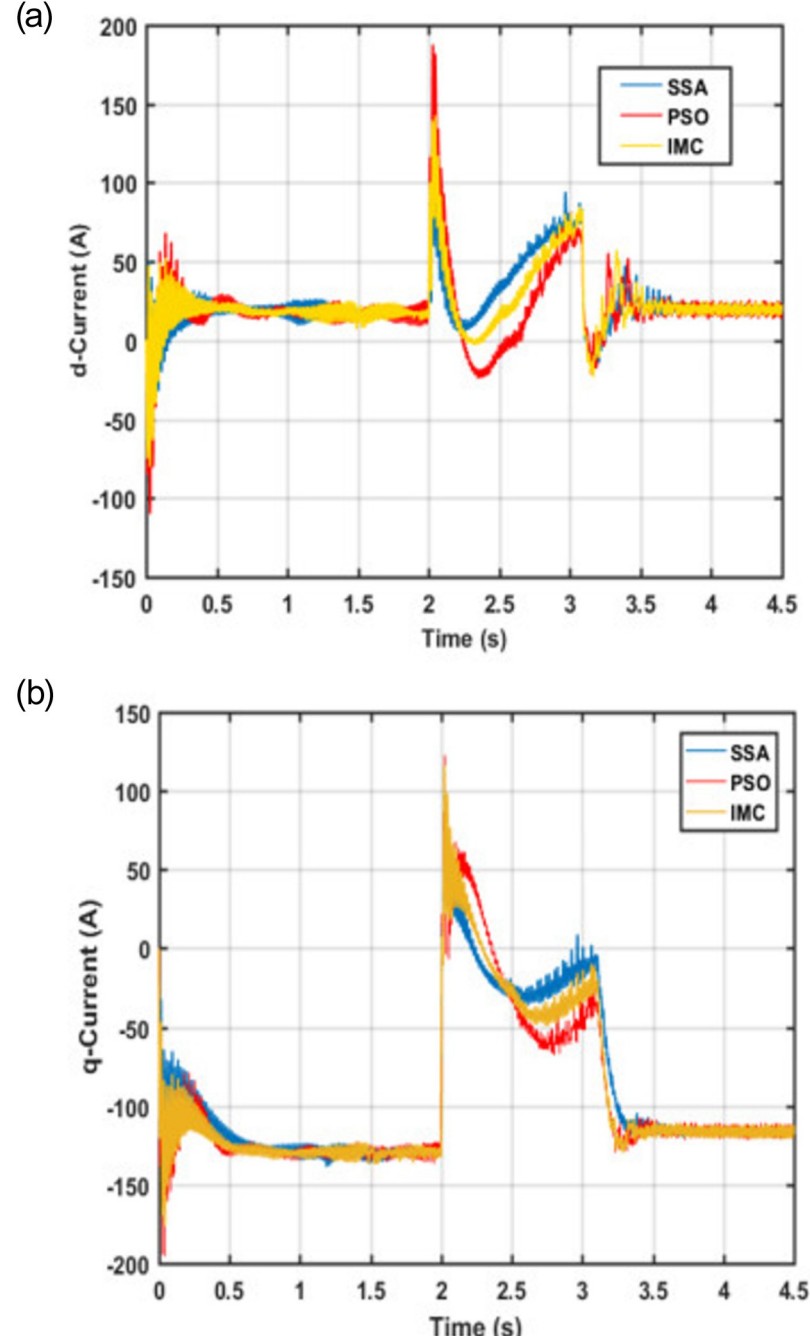

**Fig 12. CW Current (a) d-component (b) q-component.**

wind turbine control scheme is the regulation of active and reactive power. A comparative analysis of the proposed control scheme with PSO and IMC-based control schemes in terms of the active-reactive power regulation is depicted in Fig 14.

It is worthwhile to mention that the current research work makes use of an effective control approach to regulate the active and reactive power using PW current. The active power of

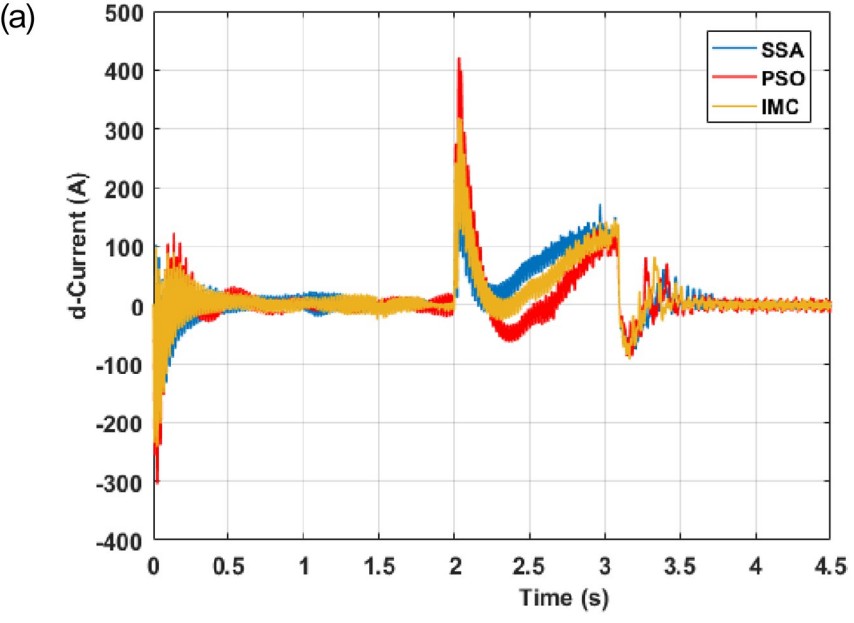

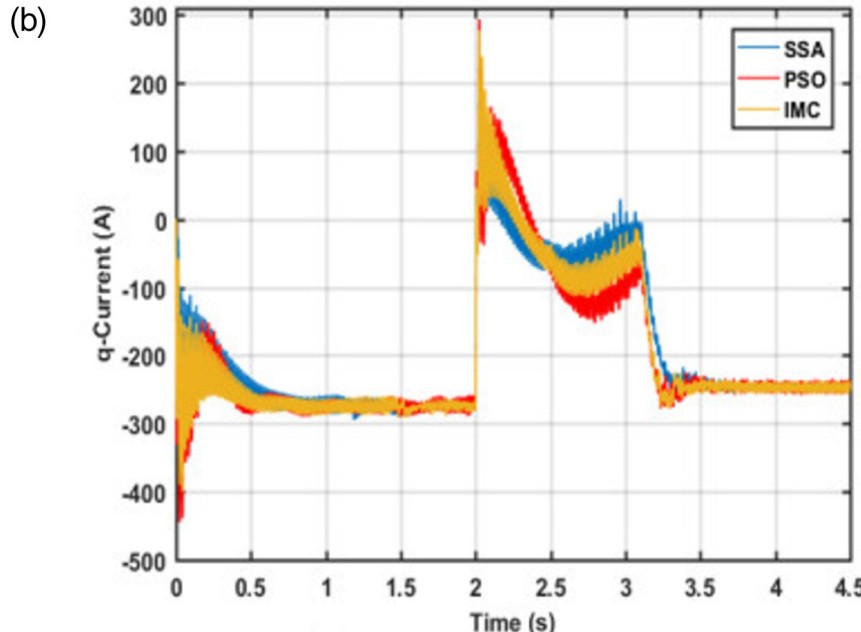

**Fig 13. PW Current (a)** *d*-**component (b)** *q*-**component.**

235kW is being supplied to the grid through PW during normal conditions as shown in Fig 14 (a). During the fault condition, the active power is reduced to almost zero magnitudes and returns to its nominal value without any noticeable oscillations and overshoots once the fault is cleared. The same is observed for reactive power as can be seen in Fig 14(b). The observed dynamic response values for the active and reactive power are shown in Table 6.

**Table 4. CW current values for d and q components.**

|  | Icd | | | Icq | | |
|---|---|---|---|---|---|---|
|  | IMC | PSO | SSA | IMC | PSO | SSA |
| tr(s) | 0.0055 | 0.0176 | 0.00204 | 0.0024 | 0.002466 | 0.002194 |
| ts(s) | 1.798 | 1.799 | 1.779 | 1.7992 | 1.7934 | 1.434 |
| os(%) | 68.654 | 89.095 | 64.095 | 37.209 | 64.739 | 41.892 |
| us(%) | 6.858 | 21.5 | 5.265 | 1.518 | 1.974 | 1.112 |
| pv(A) | 144.09 | 187.61 | 124.891 | 175.88 | 195.136 | 159.2728 |

**Table 5. PW current values for _d_ and _q_ components.**

|  | Ipd | | | Ipq | | |
|---|---|---|---|---|---|---|
|  | IMC | PSO | SSA | IMC | PSO | SSA |
| tr(s) | 0.00394 | 0.0137 | 0.00274 | 0.00286 | 0.0018 | 0.0021 |
| ts(s) | 1.798 | 1.799 | 1.786 | 1.799 | 1.7797 | 1.4546 |
| os(%) | 67.546 | 81.51 | 56.178 | 31.181 | 88.950 | 31.043 |
| us(%) | 10.073 | 14.129 | 3.741 | 1.843 | 1.880 | 1.600 |
| pv(A) | 320.6 | 421.58 | 286.68 | 398.805 | 444.8337 | 363.410 |

One of the most important parameters that need to be regulated is the dc-link voltage. Generally, it is regulated through a grid side converter. In current work, the MSC is utilized to regulate the active power, reactive power, and dc-link voltage through SSA-optimized GSC to achieve the coordination between MSC and GSC. The comparative analysis of the studied control techniques in regulating the dc-bus voltage is depicted in Fig 15.

Fig 15 shows that the SSA optimization technique performs better during both normal and fault conditions with an improved dynamic response. Following Table 7 shows the corresponding values for the dynamic response for the dc-link voltage.

## 5. Conclusion

This paper presented a flux control technique for the RSC of a BDFIG to inject the required reactive current to the grid by decreasing flux value to zero instantly and restrict the machine from developing electro motive force. The BDFIG is represented by an equivalent circuit and the single-phase equivalent circuit seen from the RSC is used to analyze the steady state components of the flux. A vector control scheme for low voltage ride through under fault conditions in grid-connected BDFIG based wind turbines is presented in this study. Using a crowbar-less approach, the proposed technique improves the dynamic response of the system by optimally tuning PI regulators through the SSA method. During the fault condition, the results shows that RSC provides the required reactive power to the grid by with analytically calculated flux. Simulation outcomes show that the BDFIG successfully rides through fault conditions. To validate the effectiveness of the proposed BDFIG vector control scheme, its dynamic response is evaluated and compared with that of the same with PSO and IMC-based controllers under identical system configuration and operating conditions. The proposed SSA-based BDFIG control scheme provides the most optimal dynamic response among the studied control schemes; thus, validates the effectiveness of the proposed control strategy. The current study has explored the effect of the optimized PI controller with symmetrical faults on the

(a)

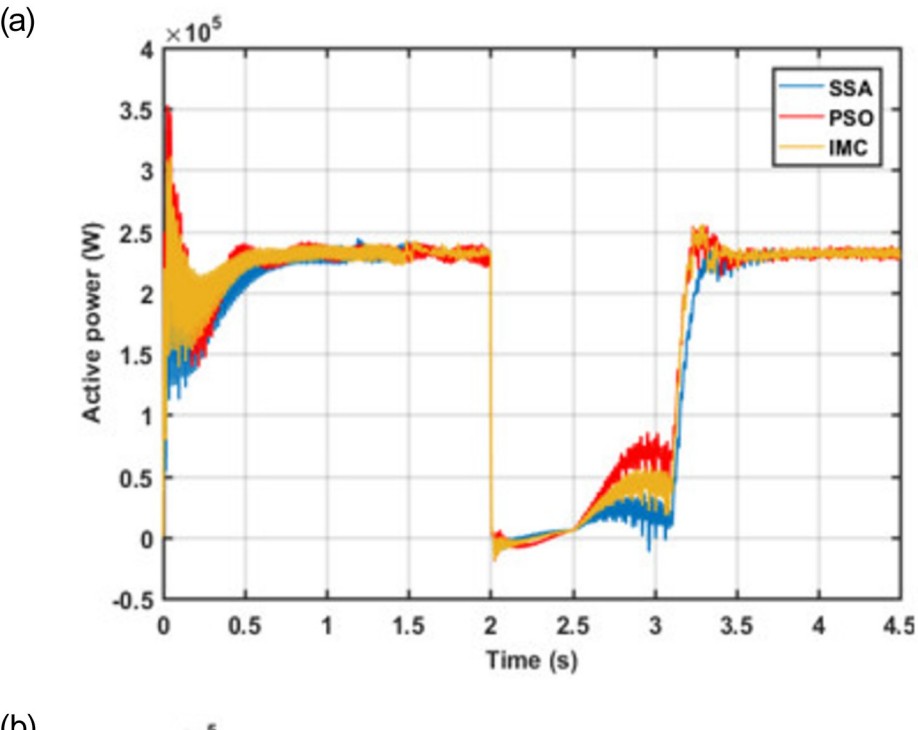

(b)

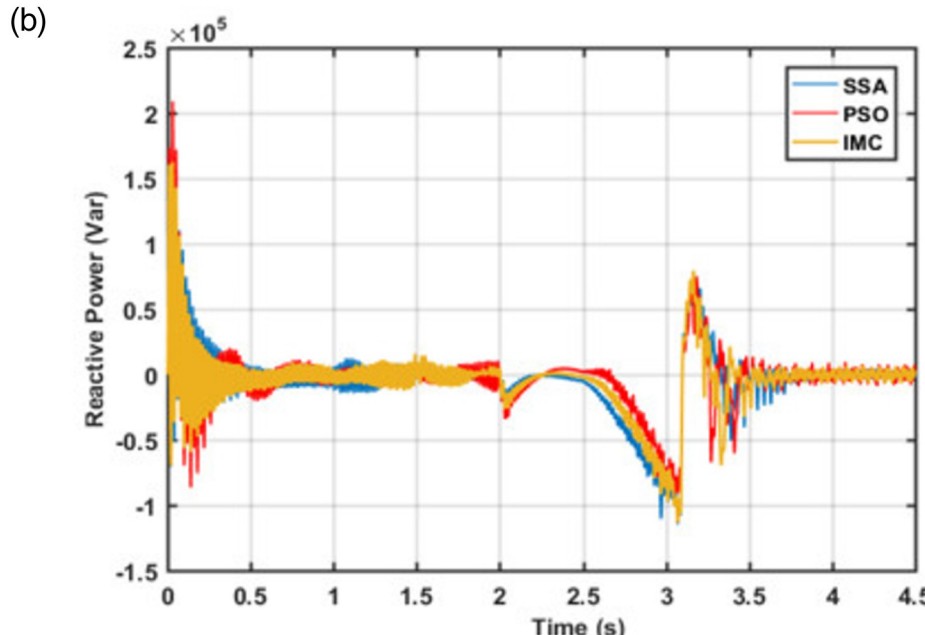

**Fig 14. (a) Active Power Response (b) Reactive Power Response.**

dynamic response of the system. The same optimization method can be explored for asymmetrical faults with different converter compositions. In addition, the nonlinear controllers may be explored and compared with the optimized PI controllers under identical operating conditions and system configurations to suggest the most efficient and suitable controller for the BDFIG based wind turbines.

**Table 6. Active and reactive power.**

|  | Active Power (Pp) | | | Reactive Power (Qp) | | |
|---|---|---|---|---|---|---|
|  | IMC | PSO | SSA | IMC | PSO | SSA |
| *tr(s)* | 0.176 | 0.853 | 0.210 | 0.009 | 0.008 | 0.006 |
| *ts(s)* | 1.798 | 1.566 | 1.798 | 1.799 | 1.799 | 1.788 |
| *os(%)* | 21.208 | 23.215 | 11.026 | 141.71 | 135.54 | 103.69 |
| *us(%)* | 22.560 | 25.560 | 31.479 | 1.196 | 0.915 | 2.03 |
| *pv(w &Var)* | 3.10e5 | 3.53e5 | 2.79e5 | 1.625e5 | 2.088e5 | 1.524e5 |

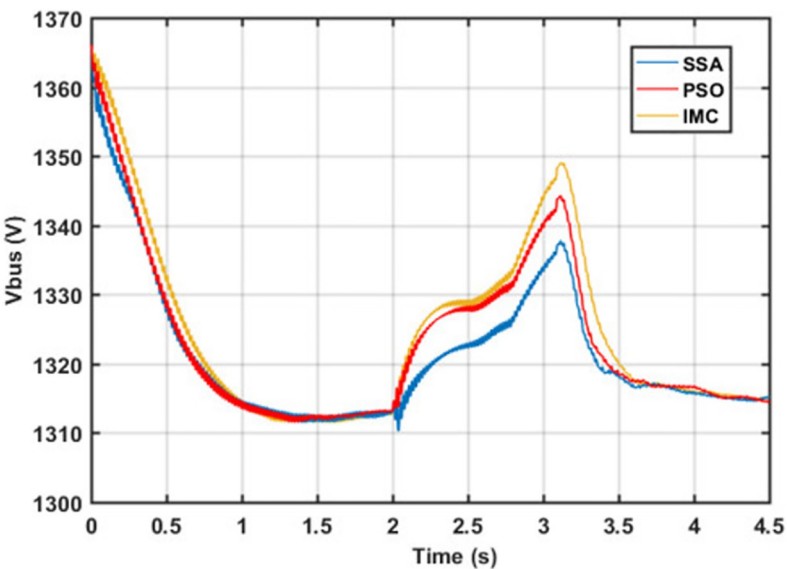

**Fig 15. DC bus voltage response.**

**Table 7. DC link voltage evaluation.**

|  | IMC | PSO | SSA |
|---|---|---|---|
| *tr*(s) | 0.000328 | 0.000295 | 0.000263 |
| *ts*(s) | 1.7275 | 1.685 | 1.558 |
| *os*(%) | 3.536 | 2.314 | 1.183 |
| *us*(%) | 4.126 | 3.265 | 2.00 |
| *pv*(V) | 1375 | 1366 | 1371 |

## Author Contributions

**Conceptualization:** Ahsanullah Memon, Mohd Wazir Bin Mustafa, Waqas Anjum, Touqeer Ahmed Jumani.

**Data curation:** Ahsanullah Memon, Saleh Masoud Abdallah Altbawi.

**Formal analysis:** Shafi Ullah, Touqeer Ahmed Jumani.

**Funding acquisition:** Shafi Ullah, Ilyas Khan, Nawaf N. Hamadneh.

**Investigation:** Ahsan Ahmed, Saleh Masoud Abdallah Altbawi.

**Methodology:** Mohd Wazir Bin Mustafa, Touqeer Ahmed Jumani.

**Project administration:** Ahsan Ahmed, Shafi Ullah, Ilyas Khan, Nawaf N. Hamadneh.

**Software:** Ahsanullah Memon, Waqas Anjum, Saleh Masoud Abdallah Altbawi.

**Supervision:** Mohd Wazir Bin Mustafa.

**Validation:** Ahsanullah Memon.

**Writing – original draft:** Mohd Wazir Bin Mustafa, Waqas Anjum.

**Writing – review & editing:** Ahsanullah Memon, Mohd Wazir Bin Mustafa, Ahsan Ahmed.

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
