## [Decision Letter · Decision Letter 0]

16 Dec 2021

PONE-D-21-35834Dynamic Response and Low Voltage Ride-Through Enhancement of Brushless Double-Fed Induction Generator Using Salp Swarm Optimization AlgorithmPLOS ONE

Dear Dr. MEMON,

Thank you for submitting your manuscript to PLOS ONE. After careful consideration, we feel that it has merit but does not fully meet PLOS ONE’s publication criteria as it currently stands. Therefore, we invite you to submit a revised version of the manuscript that addresses the points raised during the review process.

We look forward to receiving your revised manuscript.

Kind regards,

Wei Yao, Ph.D.

Academic Editor

PLOS ONE

Journal Requirements:

2. PLOS requires an ORCID iD for the corresponding author in Editorial Manager on papers submitted after December 6th, 2016. Please ensure that you have an ORCID iD and that it is validated in Editorial Manager. To do this, go to ‘Update my Information’ (in the upper left-hand corner of the main menu), and click on the Fetch/Validate link next to the ORCID field. This will take you to the ORCID site and allow you to create a new iD or authenticate a pre-existing iD in Editorial Manager. Please see the following video for instructions on linking an ORCID iD to your Editorial Manager account: https://www.youtube.com/watch?v=_xcclfuvtxQ.

Reviewers' comments:

Reviewer's Responses to Questions

**Comments to the Author**

1. Is the manuscript technically sound, and do the data support the conclusions?

Reviewer #1: Yes

Reviewer #2: Yes

2. Has the statistical analysis been performed appropriately and rigorously? 

Reviewer #1: Yes

Reviewer #2: Yes

3. Have the authors made all data underlying the findings in their manuscript fully available?

Reviewer #1: Yes

Reviewer #2: Yes

4. Is the manuscript presented in an intelligible fashion and written in standard English?

Reviewer #1: Yes

Reviewer #2: Yes

5. Review Comments to the Author

Reviewer #1: Comment 1: 

Why not directly adopt some advanced nonlinear controllers, but adopt the method of selecting optimizing PI parameters to improve its dynamic performance? The simulation results can be compared with some nonlinear controllers to show the superiority of the proposed control strategy.

Comment 2: 

The reviewer think the sentence “Despite the mentioned advantages, the investigation regarding its low voltage ride through (LVRT) is yet explicitly unattended” in the abstract is arbitrary and suggest a revision.

Comment 3: 

Note that the format includes paragraphs, references, etc., and some figures are unclear or deformed, such as Fig. 2 and Fig. 5.

Comment 4: 

How to get the "8kw" in the following paragraph of Fig. 14?

Comment 5: 

Future study should be added in Conclusion.

Reviewer #2: 1. Authors employ meta-heuristic algorithms to optimally tune PI gains of DFIG, such optimization idea and framework is a very typical optimization procedure, however, authors ignored the previous related work, hence, Introduction should be carefully rewritten to cover these important works: (a) Grouped grey wolf optimizer for maximum power point tracking of doubly-fed induction generator based wind turbine, Energy Conversion and Management.2017. (b) Democratic joint operations algorithm for optimal power extraction of PMSG based wind energy conversion system, Energy Conversion and Management. 2018. (c) Energy reshaping based passive fractional-order PID control design and implementation of a grid-connected PV inverter for MPPT using grouped grey wolf optimizer, Solar Energy, 2018. and (d) Adaptive fractional-order PID control of PMSG based wind energy conversion system for MPPT using linear observers, International Transactions on Electrical Energy Systems. 2019.

2. In case studies, more typical meta-heuristic algorithms should be compared with SSA to show its effectiveness and advantages;

3. SSA and PSO parameters should be provided;

4. Simulation environment should be given, e.g., solver type, sampling size, etc.;

5. Other advanced modification of SSA should be reviewed in Introduction, as SSA recently gains a number of interests, authors must address its applications carefully, see: Novel bio-inspired memetic salp swarm algorithm and application to MPPT for PV systems considering partial shading condition, Journal of Cleaner Production. 2019.

6. Section 2 should be significantly reduced as the presented knowledge is well-known, only provide the core contents;

7. In case studies, controller output curve should also be provided;

8. Future studies should be provided.

6. PLOS authors have the option to publish the peer review history of their article (what does this mean?). If published, this will include your full peer review and any attached files.

Reviewer #1: No

Reviewer #2: No

---

## [Author Response · Author response to Decision Letter 0]

17 Jan 2022

Dynamic Response and Low Voltage Ride-Through Enhancement of Brushless Double-Fed Induction Generator Using Salp Swarm Optimization Algorithm

We are grateful to the anonymous reviewers for their comments and suggestions. Below we respond to the comments of each reviewer in detail. We are also providing a revised manuscript that reflects their suggestions and comments. We feel that this has resulted in a stronger manuscript.

Round-01

Response to Reviewer-1

Reviewer-1, Comment -1

Why not directly adopt some advanced nonlinear controllers, but adopt the method of selecting optimizing PI parameters to improve its dynamic performance? The simulation results can be compared with some nonlinear controllers to show the superiority of the proposed control strategy.

Authors’ Response/Clarification 

Thank you for this comment. Considering the robust nature and simplicity in modeling and working mechanism, the authors have utilized the PI controller instead of a non-linear controller which is generally complex in design and comparatively sensitive to parameter variations than the PI controller. The major disadvantage of the PI controller which directly impacts the dynamic response of the system is the difficulty in its parameter selection. This research dealt with the mentioned disadvantage of PI controller by optimizing its gains automatically through a swarm intelligence-based metaheuristic technique called salp swarm optimization algorithm. Since the current is more leaned towards the optimization of the PI controller and its impact on the dynamic response of the system so we couldn’t had focused our attention towards the non-linear controllers, nevertheless, owing to this very interesting aspect of the study as mentioned by the reviewer, we have added it in our future recommendations for the researchers to explore it in greater details.

Reviewer-1, Comment -2

The reviewer think the sentence “Despite the mentioned advantages, the investigation regarding its low voltage ride through (LVRT) is yet explicitly unattended” in the abstract is arbitrary and suggest a revision

Authors’ Response

The authors completely agree with the viewpoint of the reviewer and have removed this sentence in the revised version of the manuscript.

Reviewer-1, Comment -3

Note that the format includes paragraphs, references, etc., and some figures are unclear or deformed, such as Fig. 2 and Fig. 5.

Authors’ Response/Clarification 

Thank you for this comment. All the figures have been thoroughly checked to improve their quality as suggested. 

 Reviewer-1, Comment -4

How to get the "8kw" in the following paragraph of Fig. 14?

Authors’ Response/Clarification 

Thank you for highlighting this mistake. This typo error has been rectified in the revised version of the manuscript.

Reviewer-1, Comment -5

Future study should be added in Conclusion.

Authors’ Response/Clarification 

Thank you for this important suggestion. The following recommendations have been added in the revised version of the manuscript as suggested.

“The current study has explored the effect of the optimized PI controller with symmetrical faults on the dynamic response of the system. The same optimization method can be explored for asymmetrical faults with different converter compositions. In addition, the nonlinear controllers may be explored and compared with the optimized PI controllers under identical operating conditions and system configurations to suggest the most efficient and suitable controller for the BDFIG based wind turbines”.

Round-01

Response to Reviewer-02

Reviewer-2, Comment -1

Authors employ meta-heuristic algorithms to optimally tune PI gains of DFIG, such optimization idea and framework is a very typical optimization procedure, however, authors ignored the previous related work, hence, Introduction should be carefully rewritten to cover these important works: (a) Grouped grey wolf optimizer for maximum power point tracking of doubly-fed induction generator based wind turbine, Energy Conversion and Management.2017. (b) Democratic joint operations algorithm for optimal power extraction of PMSG based wind energy conversion system, Energy Conversion and Management. 2018. (c) Energy reshaping based passive fractional-order PID control design and implementation of a grid-connected PV inverter for MPPT using grouped grey wolf optimizer, Solar Energy, 2018. and (d) Adaptive fractional-order PID control of PMSG based wind energy conversion system for MPPT using linear observers, International Transactions on Electrical Energy Systems. 2019.

Authors’ Response/Clarification 

Thank you for this comment. The authors have thoroughly studied these research works and found them very relevant to the current study. Hence all the mentioned studies have been duly added and discussed in the literature review section of the article.

Reviewer-2, Comment -2

In case of studies, more typical meta-heuristic algorithms should be compared with SSA to show its effectiveness and advantages;

Authors’ Response/Clarification 

Thank you for this comment. We had compared our results with one of the well-known metaheuristic algorithms called PSO. Since all the metaheuristic algorithms initiate their searching mechanism by generating random numbers, therefore they must be run multiple times to achieve their statistical evaluation. Thus, adding results of other algorithms along with their statistical analysis and comparing them with SSA would add complexity in terms of results interpretation and would also increase the simulation time, paper length, and less important details which may divert the core essence of the article. That’s the reason, the authors have stuck to the major goal of this study which is to explore the intelligence of SSA in achieving an optimal set of PI parameters for enhancing dynamic response and LVRT capability of the BDFIG based wind turbines which is duly achieved as evident from the outcomes of the study. 

Reviewer-2, Comment -3

SSA and PSO parameters should be provided;

Authors’ Response/Clarification 

Thank you for this comment. The utilized parameters for the PSO and SSA are added in tabular form (Table 1) in the revised version of the manuscript as suggested.

Reviewer-2, Comment -4

Simulation environment should be given, e.g., solver type, sampling size, etc.;

Authors’ Response/Clarification 

Thank you for this comment. The “Discrete Tustin/Backward Euler (TBE)” solver is used for simulating the proposed BDFIG based wind turbine model with a sampling time of 50 µs. This information regarding the solver has been added and highlighted in the revised version of the manuscript as suggested.

Reviewer-2, Comment -5

Other advanced modifications of SSA should be reviewed in the Introduction, as SSA recently gains a number of interests, authors must address its applications carefully, see: Novel bio-inspired memetic salp swarm algorithm and application to MPPT for PV systems considering partial shading condition, Journal of Cleaner Production. 2019.

Authors’ Response/Clarification 

We gladly take this suggestion. The mentioned study is very relevant to the current research work and hence added in the revised version of the manuscript as suggested.

Reviewer-2, Comment -6

Section 2 should be significantly reduced as the presented knowledge is well-known, only provide the core contents;

Authors’ Response/Clarification 

We gladly take this suggestion. The mentioned Section has been reduced accordingly.

Reviewer-2, Comment -7

In case studies, controller output curve should also be provided;

Authors’ Response/Clarification 

Thank you for this comment. The controller output in the current study is controlled pulses used to trigger the inverter. Since these pulses have been found meaningless thus their interpretation in the text would not be possible, therefore, the authors have not added them in the article. 

Reviewer-2, Comment -8

Future studies should be provided

Authors’ Response/Clarification 

Thank you for this important suggestion. The following recommendations have been added in the revised version of the manuscript as suggested.

“The current study has explored the effect of the optimized PI controller with symmetrical faults on the dynamic response of the system. The same optimization method can be explored for asymmetrical faults with different converter compositions. In addition, the nonlinear controllers may be explored and compared with the optimized PI controllers under identical operating conditions and system configurations to suggest the most efficient and suitable controller for the BDFIG based wind turbines.

---

## [Decision Letter · Decision Letter 1]

7 Mar 2022

Dynamic Response and Low Voltage Ride-Through Enhancement of Brushless Double-Fed Induction Generator Using Salp Swarm Optimization Algorithm

PONE-D-21-35834R1

Dear Dr. MEMON,

We’re pleased to inform you that your manuscript has been judged scientifically suitable for publication and will be formally accepted for publication once it meets all outstanding technical requirements.

Kind regards,

Wei Yao, Ph.D.

Academic Editor

PLOS ONE

Additional Editor Comments (optional):

Reviewers' comments:

Reviewer's Responses to Questions

**Comments to the Author**

1. If the authors have adequately addressed your comments raised in a previous round of review and you feel that this manuscript is now acceptable for publication, you may indicate that here to bypass the “Comments to the Author” section, enter your conflict of interest statement in the “Confidential to Editor” section, and submit your "Accept" recommendation.

Reviewer #1: All comments have been addressed

Reviewer #2: All comments have been addressed

2. Is the manuscript technically sound, and do the data support the conclusions?

Reviewer #1: Yes

Reviewer #2: Yes

3. Has the statistical analysis been performed appropriately and rigorously? 

Reviewer #1: Yes

Reviewer #2: Yes

4. Have the authors made all data underlying the findings in their manuscript fully available?

Reviewer #1: Yes

Reviewer #2: Yes

5. Is the manuscript presented in an intelligible fashion and written in standard English?

Reviewer #1: Yes

Reviewer #2: Yes

6. Review Comments to the Author

Reviewer #1: (No Response)

Reviewer #2: (No Response)

7. PLOS authors have the option to publish the peer review history of their article (what does this mean?). If published, this will include your full peer review and any attached files.

Reviewer #1: No

Reviewer #2: No

---

## [Editor Report · Acceptance letter]

20 Apr 2022

PONE-D-21-35834R1 

Dynamic Response and Low Voltage Ride-Through Enhancement of Brushless Double-Fed Induction Generator Using Salp Swarm Optimization Algorithm. 

Dear Dr. Memon:

I'm pleased to inform you that your manuscript has been deemed suitable for publication in PLOS ONE. Congratulations! Your manuscript is now with our production department. 

Kind regards, 

on behalf of

Professor Wei Yao 

Academic Editor

PLOS ONE